# The Characterization and Differential Analysis of m^6^A Methylation in Hycole Rabbit Muscle and Adipose Tissue and Prediction of Regulatory Mechanism about Intramuscular Fat

**DOI:** 10.3390/ani13030446

**Published:** 2023-01-28

**Authors:** Gang Luo, Yaotian Ai, Lin Yu, Shuhui Wang, Zhanjun Ren

**Affiliations:** College of Animal Science and Technology, Northwest A&F University, Xianyang 712100, China

**Keywords:** Hycole rabbit, m^6^A, gene expression, methylase, intramuscula fat

## Abstract

**Simple Summary:**

At present, one of the main problems hindering the development of rabbit industry is the low intramuscular fat of rabbits. N6-methyladenosine (m^6^A) is a highly dynamic RNA modification, which plays an important regulatory role in various physiological processes of animals. Sequencing rabbit longissimus lumborum muscle and perirenal fat by MeRIP-seq (methylated RNA immunoprecipitation with next-generation sequencing), we provided key prediction pathway for producing rabbit meat with good taste.

**Abstract:**

N6-methyladenosine (m^6^A) widely participates in various life processes of animals, including disease, memory, growth and development, etc. However, there is no report on m^6^A regulating intramuscular fat deposition in rabbits. In this study, m^6^A modification of Hycole rabbit muscle and adipose tissues were detected by MeRIP-Seq. In this case, 3 methylases and 12 genes modified by m^6^A were found to be significantly different between muscle and adipose tissues. At the same time, we found 3 methylases can regulate the expression of 12 genes in different ways and the function of 12 genes is related to fat deposition base on existing studies. 12 genes were modified by m^6^A methylase in rabbit muscle and adipose tissues. These results suggest that 3 methylases may regulate the expression of 12 genes through different pathways. In addition, the analysis of results showed that 6 of the 12 genes regulated eight signaling pathways, which regulated intramuscular fat deposition. RT-qPCR was used to validate the sequencing results and found the expression results of RT-qPCR and sequencing results are consistent. In summary, *METTL4*, *ZC3H13* and *IGF2BP2* regulated intramuscular fat by m^6^A modified gene/signaling pathways. Our work provided a new molecular basis and a new way to produce rabbit meat with good taste.

## 1. Introduction

N6-methyladenosine (m^6^A) is the most common internal modification form of mRNA in eukaryotes [1], which affects a variety of biological processes, including splicing, processing, translation, stability and decay, nuclear export, cellular metabolism and differentiation [2,3]. M^6^A is mainly regulated by methylase [4,5,6,7,8] (*ZC3H13*, *METTL4*, *KIAA1492*, *WTAP*, *METTL14*, *RBM15/15B*, *METTL3*), demethylase [9,10] (ALKBH5, FTO) and recognition enzyme (*IGF2BP2*, *YTHDF1-3*, *HNRNPA2B1*, *YTHDC1-2*) [11,12,13]. M^6^A modification is widely distributed in animal tissues such as lung, brain, kidney, liver, and heart, and exhibits tissue-specific regulation [6,14].

*METTL4* mediated m^6^A methylation based on a past phylogenetic prediction of N6-methylation [15,16,17]. Knockdown of *METTL4* led to the change of adipocyte differentiation, shown by defective gene regulation and impaired lipid production [8]. *ZC3H13* is critical for mRNA m^6^A methylation [7]. Upstream of NF-κB may be activated by *ZC3H13* [18] and NF-κB signaling plays an important role in adipocyte differentiation [19]. Studies showed *IGF2BP2* is a m^6^A methylated reading protein [20]. In addition, *IGF2BP2* is a growth factor that plays a key role in regulating the translation of *IGF2* and controls fetal growth and organ formation, including adipogenesis and pancreatic development [21]. However, the specific mechanism of *METTL4*, *ZC3H13* and *IGF2BP2* regulating fat deposition through m^6^A modification is unclear.

With the development of society, obesity has become a more and more serious health problem for contemporary people [22]. In addition, Adipose tissues plays an important role not only in heat generation and lipid storage, but also in muscle flavor. Adipose tissues deposition at the cellular level is mainly reflected by the proliferation and differentiation of adipocytes. It is generally accepted that adipocyte differentiation and proliferation are highly regulated process that involves several transcription factors and modification modes. Studies have shown that *FABP3* [23], *COL6A5* [24], *FAM13A* [25], *MYOZ2* [26,27], *EGR2* [28], *SOX9* [29], *IRX5* [30], *PRKAG 3* [31], *ABCA1* [32], *ADRB1* [33], *ADAMTS18* [34,35] and *GLI2* [36,37] are important factors in the regulation of fat deposition. M^6^A is involved in a variety of biological processes and the occurrence of a variety of diseases, including tumor, obesity and infertility [38]. Thus, it is of great value to unravel the mechanisms by which m^6^A orchestrate adipocyte differentiation and proliferation by regulatory factors.

Intramuscular fat refers to the content of fat in muscle, and the longest dorsal muscle is the main component of rabbit meat. Perirenal fat is one of the adipose tissues closest to the longest dorsal muscle, and also the main white fat that can be collected from 0-day-old rabbits. In this study, we used newborn rabbits to study the regulation of m^6^A on muscle and adipose tissues development through MeRIP-seq sequencing. Subsequently, we mapped the m^6^A map of transcriptome width of meat rabbits base on sequencing results. In addition, 3 differential methylases and 12 differential genes modified by m^6^A related to fat deposition between muscle and adipose tissues were found. 3 differential methylases regulated the expression of 12 differential genes modified by m^6^A in different ways. In addition, 6 of the 12 genes regulated eight signaling pathways, which regulate intramuscular fat deposition. The study has laid a molecular theoretical foundation for cultivating excellent rabbits.

## 2. Material and Methods

### 2.1. Ethics Statement

All experimental procedures were reviewed and approved by the Institutional Animal Care and Use Committee (IACUC) of Northwest A&F University, under the permit No. DK 2022045.

### 2.2. Animals and Tissues Collection

We collected the perirenal fat and the longissimus lumborum of 3 female Hycole rabbits. They were 0 days old and had the same environment. After rabbits were humanely sacrificed to ameliorate suffering, the longissimus lumborum and perirenal adipose tissues of rabbits were quickly separated. Subsequently, we placed the sample in the −80 °C refrigerator after quick freezing with liquid nitrogen.

### 2.3. RNA Extraction and Fragmentation

Total RNA was got by using Trizol reagent (Fankewei, Xianyang, China). We used Dynabeads^®^ mRNA purification kit (Ambion, Texas, USA) for purification after RNA detection. All of the purified mRNA was randomly broken into ∼100 nt fragments by Magnesium RNA Fragmentation Module (NEB, Beijing, China) for the construction of a MeRIP-Seq (methylated RNA immunoprecipitation and sequencing) library.

### 2.4. M^6^A Immunoprecipitation and Library Construction

The fragmented mRNA was divided into two parts, one of which used for immunoprecipitation (IP) based on the method used by Dominissini et al. [14]. The cleaved RNA fragments were combined with antibody, then reverse transcribed into cDNA. The blunt ends were connected to an A-base for ligation to the indexed adapters. After size selection, we used the heat-labile UDG enzyme (Baltimore, MD, USA) to treat the U-labeled second-stranded DNA and amplified the ligated products with PCR. Finally, we performed the 2 × 150 bp paired-end sequencing on an illumina Novaseq™ 6000 (Lianchuan, Hangzhou, China). The specific method used is the same as that of Wang et al. [39].

### 2.5. MeRIP-Seq

After cDNA synthesis, end repair, the addition of an A tail and connectors, cDNA fragment purification, and PCR enrichment of fragmented mRNA, we performed the 2 × 150 bp paired-end sequencing (PE150) on an illumina Novaseq™ 6000 (LC-Bio Technology CO., Ltd., Hangzhou, China) following the vendor’s recommended protocol.

### 2.6. RT-qPCR

Primers were designed with premier 6 software (http://www.greenxf.com/soft/190969.html (accessed on 18 January 2018)) (6.1, Premier company, Canada). Primer sequences are in Table 1. After synthesizing cDNA, we used SYBR Green [40] to conduct RT-qPCR. With β-actin as the internal reference, we use the 2^−ΔΔCt^ method to analyze the results.

### 2.7. Quality Control, Mapping and Statistical Analysis

Firstly, we used fastp software [41] to control the quality of raw reads(Remove reads that contain more than 5% of N), filtered the low-quality data (Alkali base with mass value Q ≤ 10 accounts for more than 20% of the total read) and acquire clean reads (Sequencing error rate is less than 0.01). Next, we used bowtie2 [42] to compare the high quality clean reads to the ribosomal database of the species and removed the reads of ribosomal RNA on the comparison. Comparison of reference genomes using software HISAT2 (http://daehwankimlab.github.io/hisat2) (accessed on 24 July 2020). (JOHNS HOPKINS University, Homewood, US). We made detailed statistics on the comparison of sequencing data, and evaluated the selection of reference sequences and the comparison results of sequencing data. The main analysis software of TF-motif analysis is MEME (http://meme-suite.org) (accessed on 3 September 2015) (UC San Diego, California, US) and peak between groups were analyzed by DiffBind [43]. The number of peak-related genes of GO function is to transfer peak-related genes to GO database (http://www.geneontology.org/ (accessed on 26 November 2022)) and calculate the number of peak-related genes of each term. With KEGG Pathway as the unit, hypergeometric test is applied to find out the Pathway that is significantly enriched in peak-related genes in the whole genome background. The most important biochemical metabolic pathway and signal transduction pathway involved in peak-related genes were determined through the significant enrichment of Pathway. GraphPad Prism7 (University Of California, State of California, US) was used to detect RT-qPCR and draw figures. We used two-tail Student’s t-test to analyze the significance of the different levels. The standards of significant and extremely significant are *p* < 0.05 and *p* < 0.01, respectively.

## 3. Results

### 3.1. Data Filtering and Quality Evaluation

Clean reads were obtained after removing low quality data and filtering the original data. As shown in Figure 1A,B, the sequencing data of muscle and fat are completely up to standard after processing. Cluster analysis showed that muscle and adipose tissues were significantly clustered, respectively (Figure 1C). In addition, we found that low-quality data in all organizations accounted for less than 7% of all data (including Q20% and q30%) (Appendix A). The proportion of base GC is greater than 50% after filtration (Appendix A). By comparing reads with reference sequences, we found unique mapped reads of all samples were more than 50% (Appendix A).

### 3.2. General Features of Rabbit m^6^A Methylation

The results showed that GACGGTG and GTCCATG were significantly enriched in muscle tissues and adipose tissue, which conforms the conservative sequence RRACH motif in the m^6^A peaks (Figure 2A). In addition, we found that the peaks located in the CDS and the start codons regions of muscle were more abundant than that in fat tissues whereas peaks in start codon segment and 3′UTR are more abundant in adipose tissues (Figure 2B,C). As shown in Table 2, we found 6971 peaks in muscle with an average length of 1118 bp and 7028 peaks in adipose tissues with an average length of 1117 bp. However, the genome rate of muscle tissues is less than that of adipose tissue.

### 3.3. Unique Features of Rabbit m^6^A Methylation

In this case, 5280 m^6^A peaks were in muscle and fat tissues. At the same time, we detected 1691 and 1928 peaks were specifically methylated in muscle and adipose tissues, respectively (Figure 3A). 230 peaks in muscle tissues were up-regulated and 101 peaks were down regulated compared with adipose tissues (Figure 3B). After annotating the inter group difference peak related genes, we found that 242 differential genes in muscle tissues were up-regulated and 120 differential genes were down regulated relative to adipose tissues (Figure 3C).

### 3.4. GO Analysis and KEGG Pathway Analysis of Differently Methylated Genes

The results of enrichment analysis showed that peaks related genes in adipose tissues and muscle tissues mainly regulated the processes of biological process(cellular process, response to stimulus, biological regulation, regulation of biological process, metabolic process, multicellular organismal process, positive regulation of biological process, cellular component organization or biogenesis, developmental process, signaling, localization, negative regulation of biological process), cellular component(cell, membrane part, cell part, organelle, protein containing complex, organelle part, membrane-enclosed lumen) and molecular function(binding, catalytic activity) (Figure 4A,B). In addition, peaks related genes of adipose tissues were more than those of muscle tissues in the processes of biological process, cellular component and molecular function. To predict the functions of the m^6^A-modified genes associated with fat deposition, we conducted KEGG pathway analysis. We found that the genes related to peaks mainly appeared in MAPK signaling pathway, Hippo signaling pathway, Notch signaling pathway, Wnt signaling pathway, mTOR signaling pathway, AMPK signaling pathway, cAMP signaling pathway, Adipocytokine signaling pathway, Regulation of lipolysis in adipocyte, Fatty acid metabolism and Non-alcoholic fatty liver disease pathways (Figure 5). Finally, we found that genes of signaling pathways related to fat deposition are more in adipose tissues than those in muscle tissues except MAPK signaling pathway (Figure 5).

### 3.5. Difference Analysis of Methylase and m^6^A Modified Genes between Muscle and Adipose Tissue

We identified and screened three methylases between fat and muscle tissues. As shown in Table 3, *ZC3H13*, *METTL4* and *IGF2BP2* are distributed on chromosomes 8, 9 and 14, respectively. In addition, we found 49156464 to 49163130 bp of *ZC3H13*, 59497322 to 59497522 bp of *METTL4* and 79807766 to 79979443 bp of *IGF2BP2* are modified with m^6^A. After analyzing and calculating the rpm value of methylase in the sequencing results, we found the expression levels of *METTL4* and *ZC3H13* in adipose tissues were significantly higher than those in muscle tissues whereas the expression of *IGF2BP2* in muscle tissues was significantly higher than that in adipose tissues (Figure 6A). Based on gene function, we found 12 differential expression genes associated with fat deposition between muscle and adipose tissues (Table 4). The distribution of genes on chromosomes, the location of m^6^A modified genes and their expression differences are shown in Table 4. In addition, we found the expression levels of *ABCA1*, *ADRB1* and *ADAMTS18* in adipose tissues were significantly higher than those in muscle tissues whereas the expression levels of *FABP3*, *COL6A5*, *FAM13A*, *MYOZ2*, *EGR2*, *SOX9*, *IRX5*, *PRKAG3* and *GLI2* in adipose tissues were significantly lower than those in muscle tissues (Figure 6B). To verify the sequencing results, we selected *ADRB1* gene for RT-qPCR and found *ADRB1* expression level was in adipose tissues was significantly higher than that in muscle tissues (Figure 6C).

### 3.6. Regulation of Methylase on the Expression of m^6^A Modifying Genes Related to Fat Deposition

As shown in Figure 7, we found *ZC3H13* regulated expression levels of *FAM13A*, *MYOZ2* and *ADAMTS18* genes through *NF-KB*. *METTL4* regulated expression levels of *GLI2*, *COL6A5*, *SOX9*, *FABP3* and *PRKAG3* genes through *AKT* gene. *IGF2BP2* regulated expression levels of *ABCA1*, *ADRB1*, *ADAMTS18*, *FABP3*, *COL6A5*, *FAM13A*, *MYOZ2*, *EGR2*, *SOX9*, *IRX5*, *PRKAG3* and *GLI2* genes through multiple pathways.

### 3.7. Methylase Regulated Genes Related to Fat Deposition through Signal Pathway

In order to explore the connection between m^6^A modifying genes related to fat deposition and signaling pathways, we made Sankey diagram and found that 6 genes regulated 8 signaling pathways by multiple pathways (Figure 8). *SOX9*, *GLI2*, *FABP3* and *ABCA1* can regulate cAMP, Hippo, PPAR and Fat digestion and absorption signaling pathways, respectively. *ADRB1* is involved in the regulation of cAMP and regulation of lipolysis in adipocyte signaling pathways. In addition, PPKAG3 play important roles in Adipocytokine signaling pathway, AMPK signaling pathway and Non-alcoholic fatty liver disease (NAFLD) signaling pathway (Figure 8).

## 4. Discussion

Adipose tissues can regulate energy storage and energy metabolism [44,45]. In addition, adipose tissues are the main component of intramuscular fat and intramuscular fat is the main factor affecting meat quality and flavor. Studies have shown that m^6^A plays an important role in the regulation of fat deposition. Therefore, we explored the regulatory mechanism of m^6^A on intramuscular fat deposition by MeRIP-seq sequencing of rabbit muscle and adipose tissues.

In this study, we discovered that rabbit mRNA m^6^A sites are mainly enriched around stop codons, CDS, and 3′UTRs by using MeRIP-Seq technology, which shared a distribution similar to that of pigs [46]. “RRACH” has been confirmed as the specific sequence of m^6^A motif by multiple sequencing [14,47,48,49]. However, “RRACH” sequence appeared in large numbers in our current data, which indicated the transcriptome of rabbits have multiple m^6^A modification sites. In addition, GO and KEGG pathway analysis were performed to deduce function on fat deposition of m^6^A modified genes. After GO enrichment analysis, the differential peaks related genes are mainly distributed in biological process (cellular process, biological regulation, metabolic process, regulation of biological process, response to stimulus, multicellular organismal process, positive regulation of biological process, cellular component organization or biogenesis, developmental process, signaling, localization, negative regulation of biological process), which are the main process of fat deposition (Figure 4). The results of KEGG pathway analysis showed main signal pathways play a vital role in the regulation of fat deposition, such as MAPK signaling pathway [50], Hippo signaling pathway [51], Notch signaling pathway [52], Wnt signaling pathway [53], mTOR signaling pathway [54], AMPK signaling pathway [55], cAMP signaling pathway [56], Adipocytokine signaling pathway, Regulation of lipolysis in adipocyte, Fatty acid metabolism and Non-alcoholic fatty liver disease pathways (Figure. 5). The results indicated that m^6^A regulates fat deposition through multiple signaling pathways.

In the present study, the expression levels of *METTL4* and *ZC3H13* in adipose tissues were significantly higher than those in muscle tissues whereas the expression levels of *IGF2BP2* in adipose tissues was significantly lower than that in muscle tissues. The results are consistent with previous studies that *METTL4* [8] and *ZC3H13* [18,19] promoted fat production whereas *IGF2BP2* [21] inhibited fat production. Our results showed that several important lipogenic genes, including *ABCA1*, *ADRB1*, *ADAMTS18*, *FABP3*, *COL6A5*, *FAM13A*, *MYOZ2*, *EGR2*, *SOX9*, *IRX5*, *PRKAG3* and *GLI2* had variations in both m^6^A methylation and mRNA expression. As an important part of fat deposition genetic network, *METTL4*, *ZC3H13* and *IGF2BP2* can regulate the expression levels of differential genes through multiple pathways. *IGF2BP2* inhibited *PID1* expression through the *DANCR*/*FOXO1* axis [57] and *FOXO1* played an important role in regulating Wnt/β-catenin signaling [58]. As the upstream target of *IRX5*, WNT/Rspo1/β-catenin regulates the expression of *IRX5* [59]. Secondly, *IGF2BP2* regulates macrophage phenotypic activation by stabilizing *PPARγ* [60]. A *PPARγ*-*LXR*-*ABCA1* pathway played an important role in cholesterol efflux and atherogenesis [61]. Next, *IGF2BP2* regulates the expression level of miR-224-5p through interacting m^6^A/*FOXM1* manner [62,63]. *EGR2* is a target gene of miR-224-5p [64]. Finally, *IGF2BP2* affected first-phase insulin secretion during hyperglycemic clamps [65]. On the one hand, insulin can regulate the expression of *ADRB1* through *STAT3*/ miR-19a pathway [66,67,68]. On the other hand, insulin stimulates translocation of *NF-kB* to the nucleus by activating a *PI3K*/*AKT*/*P70S6*/*p38-MAPK*/*MAPKAPK-2* cascade [69].

Another methylase *ZC3H13* is responsible for activating *NF-κB* [18]. Curcumin down-regulated *ADAMTS18* gene though *NF-κB* Signaling Pathway [70]. *NF-κB* can regulate the activity of *MYOZ2* gene [71]. In addition, *NF-κB* played critical roles in the regulation of *IL-1β* [72] and *IL-1β* in adipose tissues increased inhibition of *FAM13A* expression [73]. In addition, knockdown of *METTL4* led to decreased N6-adenine methylation at the promoter region of *INSR* gene, down-regulated gene expression, inactivated the *INSR* pathway, and promotes lipid production of adipocytes [8]. *INSR* regulates type 2 diabetic through *IRS-1*/*PI3K* pathway [74]. *PI3K* played an important role in protecting intestinal stem cells through the *AKT*/p53 axis [75]. *P53* activates miR-192-5p to mediate *FABP3* gene [76,77]. Second, *AKT* regulates cancer cells by *PAK1*/*ATF2*/miR-132 signaling axis [78,79] and *PRKAG3* is a target gene of miR-132 [80]. Third, *AKT* Regulates Mitochondrial Biogenesis by *GSK-3β*/*PGC-1α* Pathway [81] and *PGC-1a* directly interacts with *SOX9* and promotes *SOX9*-dependent transcriptional activity [82]. Fourth, *AKT* plays an important role in regulating miR-455-5p [83] and *USP3* is the target gene of miR-455-5p [84]. *USP3* regulates *COL9A3*/*COL6A5* stabilisation to promote gastric cancer progression [85]. Finally, *AKT* stimulates translocation of *NF-kB* to the nucleus by *P70S6*/*p38-MAPK*/*MAPKAPK-2* cascade [69] and *p38 MAPK* Signal Pathway can inhibit miR-1 expression [86]. MiR-1 can delay the progression of osteoarthritis by inhibiting *IHH* [87] and *IHH* regulates *RUNX2* through *GLI2* to stimulate osteoblast differentiation [88]. These results indicated *METTL4*, *ZC3H13* and *IGF2BP2* can regulate the expression of *ABCA1*, *ADRB1*, *ADAMTS18*, *FABP3*, *COL6A5*, *FAM13A*, *MYOZ2*, *EGR2*, *SOX9*, *IRX5*, *PRKAG3* and *GLI2* genes through multiple pathways.

The results of Sankey diagram show that 6 m^6^A-modified genes regulated cAMP signaling pathway, Hippo signaling pathway, PPAR signaling pathway, Fat digestion and absorption signaling pathway, Regulation of lipolysis in adipocyte signaling pathways, Adipocytokine signaling pathway, AMPK signaling pathway and Non-alcoholic fatty liver disease (NAFLD) signaling pathway. Studies indicated cAMP regulates the fat deposition in adipose tissues of finishing pigs through PKA pathway regulates [89]. Hippo and PPAR signaling pathways regulate adipocyte differentiation [90,91]. AMPK suppresses the adipogenesis in human adipose derived stem cells [92]. Fat digestion and absorption signaling pathway, Regulation of lipolysis in adipocyte signaling pathways, Adipocytokine signaling pathway and Non-alcoholic fatty liver disease (NAFLD) signaling pathway are classical signaling pathways regulating fat deposition in different tissues. These results indicated 8 signaling pathways can regulate fat deposition. At the same time, 8 signaling pathways are present in the muscle tissue. Therefore, cAMP signaling pathway, Hippo signaling pathway, PPAR signaling pathway, Fat digestion and absorption signaling pathway, Regulation of lipolysis in adipocyte signaling pathways, Adipocytokine signaling pathway, AMPK signaling pathway and Non-alcoholic fatty liver disease (NAFLD) signaling pathway are the main signal pathway regulating intramuscular fat. Although we have carried out a significant amount of work, there are still many deficiencies. In order to further explore the mechanism of intramuscular fat deposition, we will later verify these regulatory mechanisms in cells and in vivo to cultivate excellent rabbit breeds.

## 5. Conclusions

In summary, we revealed features of m^6^A distribution in the Hycole rabbit perirenal fat and longissimus lumborum. Next, we screened 3 differential methylases and 12 genes modified by m^6^A between muscle and adipose tissues. Based on the previous studies, we found that three methylases can regulate the expression of m^6^A modified genes through different pathways. In addition, 6 of the 12 genes regulated eight signaling pathways, which regulated fat deposition in muscle tissue. These results indicated 3 differential methylases regulated intramuscular fat deposition via m^6^A modifying gene/signaling pathways. This study provided a solid theoretical foundation and a new way to produce rabbit meat with good taste.

## Figures and Tables

**Figure 1 animals-13-00446-f001:**
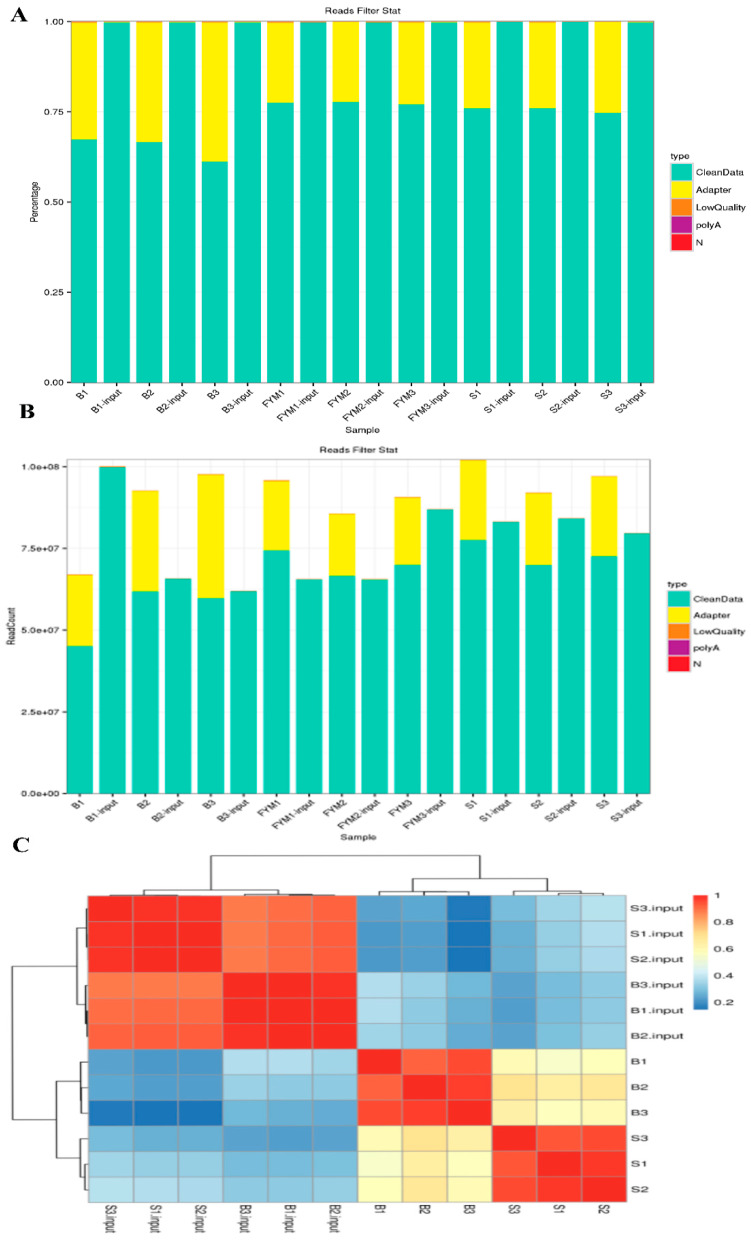
Data processing and sample clustering; (**A**) Data preprocessing distribution (percentage); (**B**) Data preprocessing distribution(value); (**C**) Clustering of muscle and fat samples. (‘B’ refers to the longest dorsal muscle and ‘S’ refers to perirenal fat).

**Figure 2 animals-13-00446-f002:**
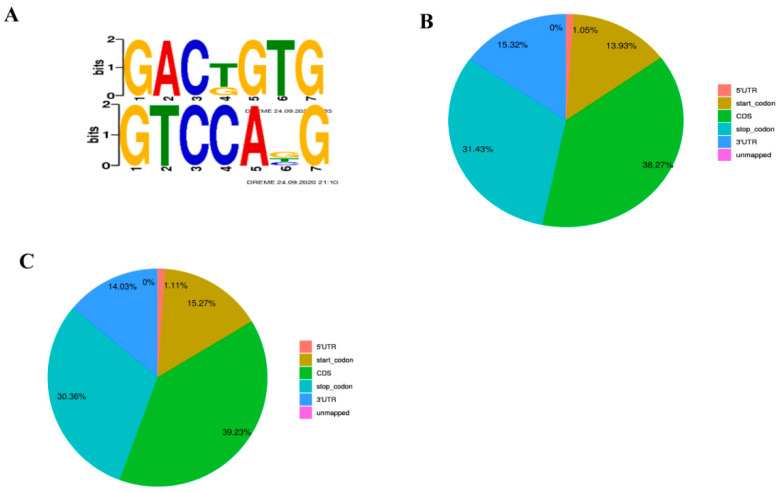
Summary of peaks (**A**) Motif sequence diagram; (**B**) Distribution of peaks in adipose tissue; (**C**) Distribution of peaks in muscle tissue. (‘B’ refers to the longest dorsal muscle and ‘S’ refers to perirenal fat).

**Figure 3 animals-13-00446-f003:**
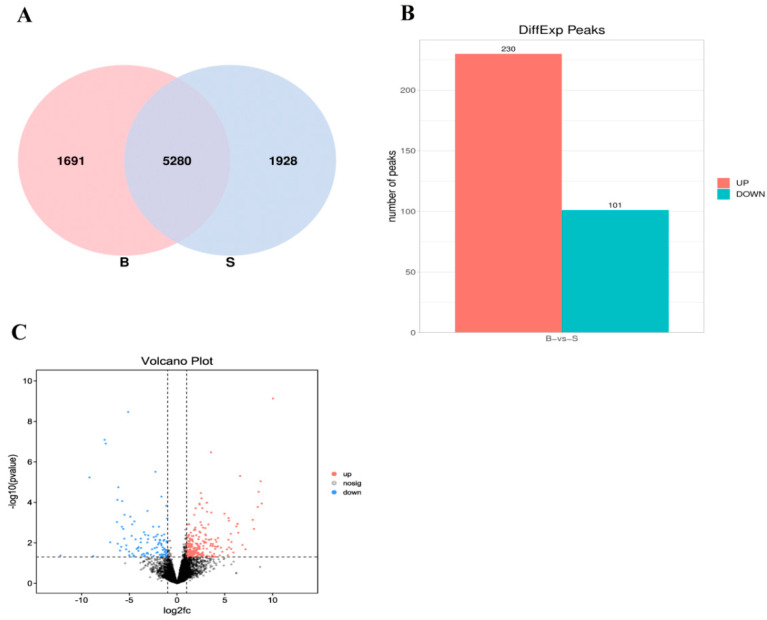
Peak analysis of difference between groups; (**A**) Common and unique peak of adipose and muscle tissue; (**B**) Difference peak histogram; (**C**) Analysis of peak related genes between adipose and muscle tissue. (‘B’ refers to the longest dorsal muscle and ‘S’ refers to perirenal fat).

**Figure 4 animals-13-00446-f004:**
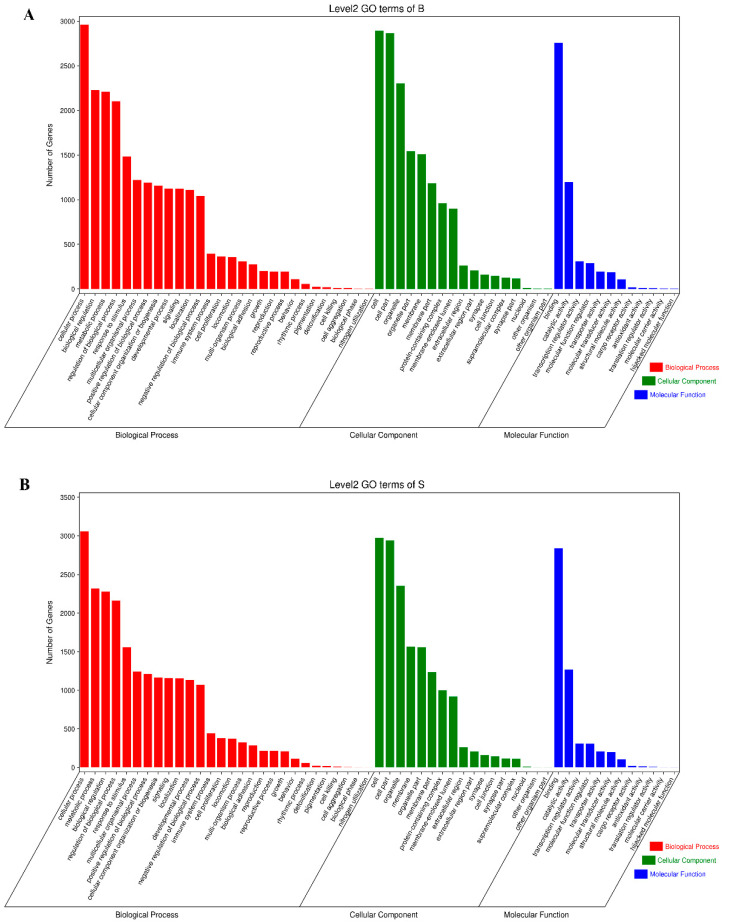
Go enrichment analysis of differential peak related genes; (**A**) Go enrichment classification histogram of muscle tissue; (**B**) Go enrichment classification histogram of fat tissue.

**Figure 5 animals-13-00446-f005:**
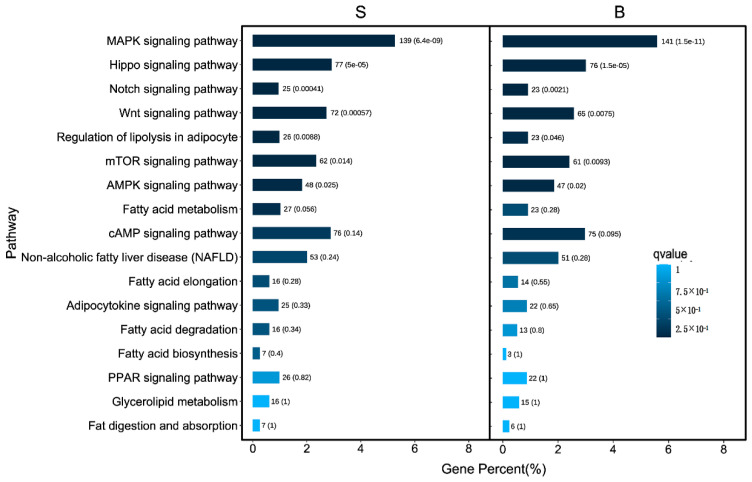
Enrichment analysis of fat metabolism related pathway KEGG. (‘B’ refers to the longest dorsal muscle and ‘S’ refers to perirenal fat). Note: The pathway is the vertical coordinate, the number of all peak-related genes are indicated on the horizontal coordinate.

**Figure 6 animals-13-00446-f006:**
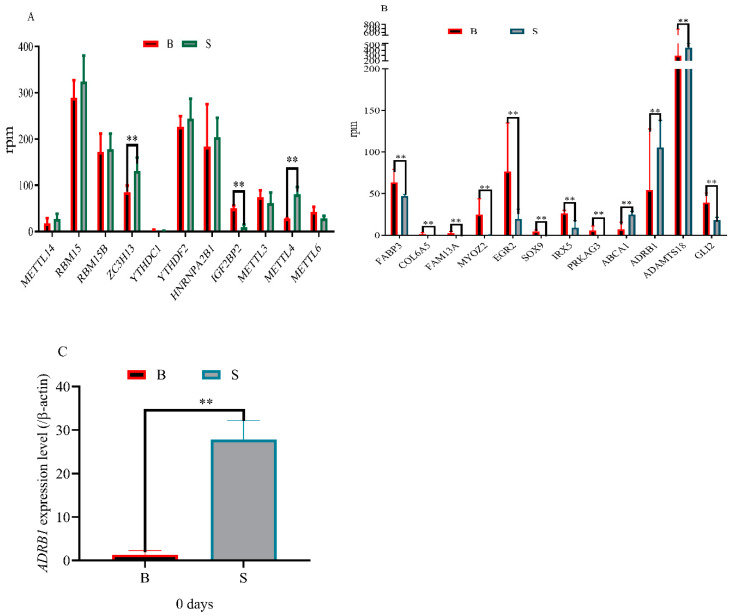
Overview of differentially expressed of methylase genes and key genes in muscle and adipose tissues samples (**A**) RPM of the methylase genes in muscle and adipose tissues; (**B**) RPM of the key genes in muscle and adipose tissues; (**C**) Expression level of *ADRB1* gene in muscle and adipose tissues (*p* < 0.01). (‘B’ refers to the longest dorsal muscle and ‘S’ refers to perirenal fat) (“**”, *p* ≤ 0.01).

**Figure 7 animals-13-00446-f007:**
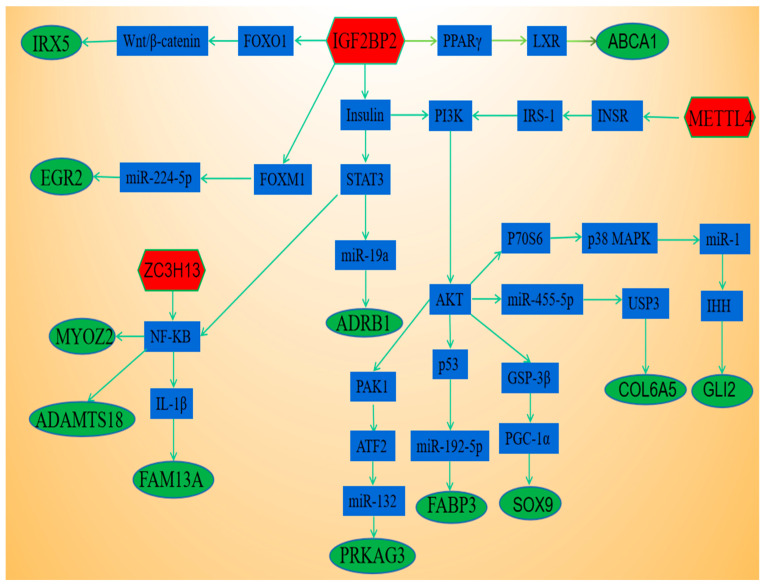
Regulation of fat related genes by methylase.

**Figure 8 animals-13-00446-f008:**
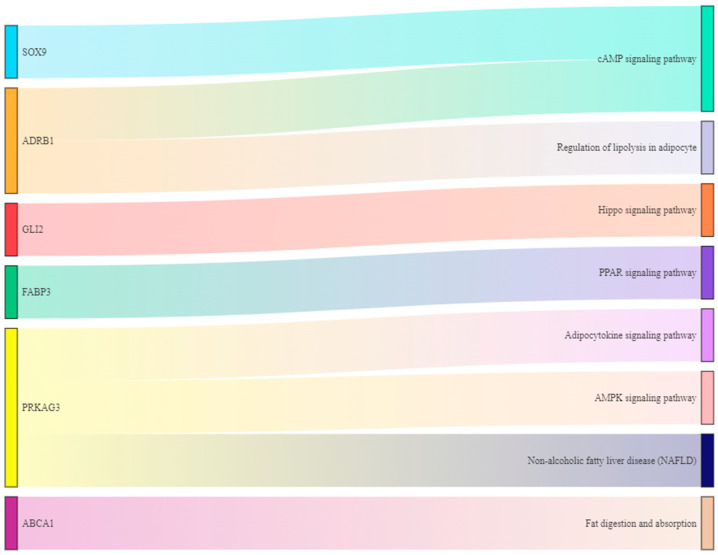
Association between genes and signaling pathways.

**Table 1 animals-13-00446-t001:** Primers used in this study.

Gene Name	Primer Sequence (5′→3′)	(Tm/°C)
*β-actin*	GGAGATCGTGCGGGACAT	61.4
	GTTGAAGGTGGTCTCGTGGAT	
*ADRB1*	CATCATCATGGGCGTGTTCA	51.8
	TAGCCGAGCCAGTTGAAGAA	

**Table 2 animals-13-00446-t002:** Summary of peaks.

Sample	Peak Number	Total Length	Average Length	Genome Rate (%)
B	6971	7800148	1118	0.347
S	7028	8055673	1117	0.358

Note: ‘B’ refers to the longest dorsal muscle and ‘S’ refers to perirenal fat.

**Table 3 animals-13-00446-t003:** Summary of methylase genes between muscle and fat tissues.

Symbol	Chromosome	Start	End	Gene_Start	Gene_End
ZC3H13	8	49156464	49163130	49112198	49203390
METTL4	9	59497322	59497522	59489716	59522894
IGF2BP2	14	79807766	79979443	79807317	79979658

**Table 4 animals-13-00446-t004:** Difference analysis of fat related genes between muscle and fat tissues.

Symbol	Chromosome	Gene_Start	Gene_End	Diff.log2FC	*p*-Value
*ABCA1*	1	8128276	8262632	1.258322149	0.030511101
*IRX5*	5	11211307	11214739	3.145191738	0.043951603
*ADAMTS18*	5	31919264	32078296	1.763515547	0.026105306
*GLI2*	7	59796306	59899847	2.106650174	0.018530687
*PRKAG3*	7	160092580	160099205	−3.456695972	0.047136086
*FABP3*	13	134849947	134860535	−1.076952166	0.008346079
*COL6A5*	14	963915	1100656	−7.002296213	0.009463418
*FAM13A*	15	60958616	61325685	−5.001719634	0.041259027
*MYOZ2*	15	94465177	94502456	3.759079992	0.011579319
*EGR2*	18	23380925	23387584	1.671242666	0.03208062
*ADRB1*	18	61629975	61631390	5.667263578	0.020071817
*SOX9*	19	55820444	55823755	−5.772627603	0.001606224

## Data Availability

Raw data can be obtained from NCBI database. The submission ID is SUB11353640 and BioProject ID is PRJNA830993.

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
