# Peer review of "The Characterization and Differential Analysis of m6A Methylation in Hycole Rabbit Muscle and Adipose Tissue and Prediction of Regulatory Mechanism about Intramuscular Fat"

_animals, 2023, doi:10.3390/ani13030446_

Round 1

Reviewer 1 Report

The manuscript addresses a question important for rabbit breaders,namely creating a solid basis to improve intramuscular fat deposition. Based on the published information they performed a targeted analysis and the obtained data were analysed from multiple aspects. The results line up with earlier data obtained in swine.The conclusions are correct and could be stimulating to alter intramuscular fat deposition.

Author Response

The manuscript addresses a question important for rabbit breaders,namely creating a solid basis to improve intramuscular fat deposition. Based on the published information they performed a targeted analysis and the obtained data were analysed from multiple aspects. The results line up with earlier data obtained in swine.The conclusions are correct and could be stimulating to alter intramuscular fat deposition.

Thank you very much for your guidance to my manuscript.

Reviewer 2 Report

The objective of the manuscript called “Functional identification of m6A in Hycole rabbit muscle and adipose tissue and prediction of methylase–mRNA-signaling pathway regulated intramuscular fat deposition pathways” was to study the regulation of m6A on muscle and adipose tissues development through MeRIP-seq sequencing

This manuscript include a very interested study that associate epigenetic and expression data. However, the summary and several parts of the manuscript are confusing and needs to be rewritten. For example, the objective of the study is not clear in the summary.

Minor changes

Delated the last sentence of the introduction and conclusion sections.

Line 78, include the supplier of the Trizol reagent

Line 97, include the complete reference of the software premier 6.

Line 98, include the reference of the SYBR Green.

Section 2.6. RT-qPCR, why the authors select these genes?

2.7. Quality control, mapping, and statistical analysis, please provide more details about these analyses.

Table 2 and 3 could be supplementary materials.

Author Response

Thank you very much for your guidance to my manuscript. We have made relevant revisions according to the proposed guidance. I hope to be approved by editor and reviewer.

The specific reply is as follows:

The objective of the manuscript called “Functional identification of m6A in Hycole rabbit muscle and adipose tissue and prediction of methylase–mRNA-signaling pathway regulated intramuscular fat deposition pathways” was to study the regulation of m6A on muscle and adipose tissues development through MeRIP-seq sequencing This manuscript include a very interested study that associate epigenetic and expression data. However, the summary and several parts of the manuscript are confusing and needs to be rewritten. For example, the objective of the study is not clear in the summary.

Minor changes

Delated the last sentence of the introduction and conclusion sections.

We have corrected it

Line 78, include the supplier of the Trizol reagent

We have corrected it

Line 97, include the complete reference of the software premier 6.

There is no reference on the development of software premier 6, but we have added the software download website.

Line 98, include the reference of the SYBR Green.

We have added it

Section 2.6. RT-qPCR, why the authors select these genes?

First of all, the gene is expressed in both fat and muscle tissues, and the expression difference is very significant. Secondly, the gene plays an important role in fat deposition. Then, the expression of this gene in fat is significantly higher than that in muscle tissue, which indicates that it may regulate adipocytes in muscle tissue and adipose tissue.

2.7. Quality control, mapping, and statistical analysis, please provide more details about these analyses.

We have corrected it

Table 2 and 3 could be supplementary materials.

We have corrected it

Reviewer 3 Report

Comments to the author:

In this study, the authors performed MeRIP-Seq on muscle and adipose tissue from day 0 Hycole rabbits. Three methylases and 12 genes modified by m6A were found to be significantly different between different tissues. The article's theme is ambiguous.

The specific issues and suggestions in the article are as follows:

1. The article title should be shorter.

2. In the Introduction's third paragraph, what is the necessary connection between obesity and the topic of this article?

3. The introduction should properly explain why longissimus muscle and perirenal adipose tissues were selected as the research objects for intramuscular fat.

4. Why did the author choose day 0 instead of adult rabbits? Was there intramuscular fat deposition at day 0?

5. What does the sample name represent in table 2?

6. In table 5, what is the strategy of methylases screening in different tissues?

7. In result 3.5, Why only the ADRB1 gene is selected to validate sequencing results?

8. In figure 6A, please add the significance of METTL4 gene.

9. The characterization and differential analysis of m6A methylation in rabbit muscle and adipose tissue is more consistent with the main content of this article.

10. In the results section, please increase the resolution of all figures to visualize these results better.

11. There are some verbal errors throughout the manuscript.

Author Response

Thank you very much for your guidance to my manuscript. We have made relevant revisions according to the proposed guidance. I hope to be approved by editor and reviewer.

The specific reply is as follows:

Comments to the author:

In this study, the authors performed MeRIP-Seq on muscle and adipose tissue from day 0 Hycole rabbits. Three methylases and 12 genes modified by m6A were found to be significantly different between different tissues. The article's theme is ambiguous.

The specific issues and suggestions in the article are as follows:

  1. The article title should be shorter.

We have corrected it

  1. In the Introduction's third paragraph, what is the necessary connection between obesity and the topic of this article?

Excessive fat deposition will lead to obesity. Both rabbits and humans are mammals, studying the fat deposition of rabbits can provide a reference for human obesity.

  1. The introduction should properly explain why longissimus muscle and perirenal adipose tissues were selected as the research objects for intramuscular fat.

We have corrected it

4 Why did the author choose day 0 instead of adult rabbits? Was there intramuscular fat deposition at day 0?

First of all, 0-day-old is the beginning of life in the external environment, and all organs and tissues have developed completely. At the same time, the regulatory function of the gene has emerged. In addition,the modification of m6A is a reversible process, which will be affected by external factors. The 0-day-old rabbits were not affected by the environment, and the genetic control of traits was dominant. Using 0 day old rabbits as experimental materials is more beneficial to the later breeding.

  1. What does the sample name represent in table 2?

We have marked the sample name

  1. In table 5, what is the strategy of methylases screening in different tissues?

First of all, only dozens of methylases have been found so far. As shown in Figure 6a, only 11 of them regulate the development of muscle and fat in the sequencing results. Finally, it is found that there are only 3 significant differences in m6A modification between fat and muscle tissues, as shown in Table 5.

  1. In result 3.5, Why only the ADRB1 gene is selected to validate sequencing results?

First of all, the gene is expressed in both fat and muscle tissues, and the expression difference is very significant. Secondly, the gene plays an important role in fat deposition. Then, the expression of this gene in fat is significantly higher than that in muscle tissue, which indicates that it may regulate adipocytes in muscle tissue and adipose tissue.

  1. In figure 6A, please add the significance of METTL4 gene.

We have corrected it

  1. The characterization and differential analysis of m6A methylation in rabbit muscle and adipose tissue is more consistent with the main content of this article.

We have corrected it

10 In the results section, please increase the resolution of all figures to visualize these results better.

We have corrected it

11 There are some verbal errors throughout the manuscript.

We have corrected it

Reviewer 4 Report

In this study, Luo et al. performed methylated RNA immunoprecipitation with next generation sequencing and detected the m6A transcriptome-wide map in longissimus lumborum and perirenal adipose tissues of Hycole rabbit. Their findings suggest that 3 methylases and 12 genes modified by m6A are significantly different between muscle and adipose tissues. More specifically, METTL4, ZC3H13, and IGF2BP2 regulated intramuscular fat by m6A-modified gene/signaling pathways. This is an interesting and informative study that sheds light on the regulatory role of the m6A modification in mammals, and especially on intramuscular fat deposition in rabbits. However, there are several issues in the manuscript that need to be addressed:

Major issues:

1.     The Authors state in several parts of the manuscript that their study proposes a new way to produce rabbit meat with good taste. However, there is no prevalent way suggested in the text; which alteration or interference in the production manner is exactly proposed?

2.     Additional details could be provided in the Materials and methods section, for instance regarding the construction of the MeRIP-Seq library.

3.     Which criteria were implemented for the filtering of low-quality data in order to get clean reads?

4.     The Discussion could be largely improved. It is mainly a repetition of the results, and especially the third paragraph seems like an extensive apposition of interactions and pathways and is hard to follow. The Authors are advised to include their in-depth commentary on their findings, as well as the limitations and future perspectives of this study.

Minor issues:

5.     The Introduction could provide more specific background information on the molecular basis of this study. For example, what are some specific effects of the N6-methyladenosine (m6A) modification?

6.     The GO analysis and KEGG pathway analysis should also be described in the materials and methods, since they are part of the workflow.

7.     The font of all Figures, and especially Figures 1 and 4, is too small and cannot be easily read. In addition, the legend of Figure 6 should contain what do the “B” and “S” refer to.

8.     Proofreading of the manuscript is needed since there are a few syntax and grammar errors.

Author Response

Thank you very much for your guidance to my manuscript. We have made relevant revisions according to the proposed guidance. I hope to be approved by editor and reviewer.

The specific reply is as follows:

In this study, Luo et al. performed methylated RNA immunoprecipitation with next generation sequencing and detected the m6A transcriptome-wide map in longissimus lumborum and perirenal adipose tissues of Hycole rabbit. Their findings suggest that 3 methylases and 12 genes modified by m6A are significantly different between muscle and adipose tissues. More specifically, METTL4, ZC3H13, and IGF2BP2 regulated intramuscular fat by m6A-modified gene/signaling pathways. This is an interesting and informative study that sheds light on the regulatory role of the m6A modification in mammals, and especially on intramuscular fat deposition in rabbits. However, there are several issues in the manuscript that need to be addressed:

Major issues:

  1. The Authors state in several parts of the manuscript that their study proposes a new way to produce rabbit meat with good taste. However, there is no prevalent way suggested in the text; which alteration or interference in the production manner is exactly proposed?

The study in this manuscript is to lay the foundation for subsequent research. Firstly, we detected the gene modified by m6A in muscle and fat by MeRIP-seq. Secondly, the differential genes related to fat deposition that are expressed in both muscle and adipose tissue were screened. Then,we chose the genes expressed in adipose tissue higher than those in muscle tissue to study their regulatory pathway. After confirming the gene, we will verify it on cells and living bodies.Finally, we will regulate rabbit fat deposition by editing genes.

  1. Additional details could be provided in the Materials and methods section, for instance regarding the construction of the MeRIP-Seq library.

We have added it

  1. Which criteria were implemented for the filtering of low-quality data in order to get clean reads?

We have added it

  1.     The Discussion could be largely improved. It is mainly a repetition of the results, and especially the third paragraph seems like an extensive apposition of interactions and pathways and is hard to follow. The Authors are advised to include their in-depth commentary on their findings, as well as the limitations and future perspectives of this study.

We have added it. This study has not done enough molecular validation, and the conclusions are predicted by sequencing results. In order to make the results more accurate, a large number of previous studies have been introduced.

Minor issues:

  1. The Introduction could provide more specific background information on the molecular basis of this study. For example, what are some specific effects of the N6-methyladenosine (m6A) modification?

We have added it

  1. The GO analysis and KEGG pathway analysis should also be described in the materials and methods, since they are part of the workflow.

We have added it

  1.   The font of all Figures, and especially Figures 1 and 4, is too small and cannot be easily read. In addition, the legend of Figure 6 should contain what do the “B” and “S” refer to.

We have corrected it

  1.     Proofreading of the manuscript is needed since there are a few syntax and grammar errors.

We have corrected it

Round 2

Reviewer 4 Report

The Authors have adequately addressed the Reviewers’ comments; the appropriate corrections were made and incorporated in the text and the revised manuscript is significantly improved. The clarifications that are provided contribute to the coherence and quality of the study. I would only suggest dividing the third paragraph of the Discussion into two (or more) paragraphs, since currently it is too lengthy and can be difficult to comprehend. Overall, the paper is well-written and contributes to the existing knowledge in its field.

Author Response

Thank you very much for your guidance to my manuscript. We have made relevant revisions according to the proposed guidance. I hope to be approved by editor and reviewer.

The specific reply is as follows:

The Authors have adequately addressed the Reviewers’ comments; the appropriate corrections were made and incorporated in the text and the revised manuscript is significantly improved. The clarifications that are provided contribute to the coherence and quality of the study. I would only suggest dividing the third paragraph of the Discussion into two (or more) paragraphs, since currently it is too lengthy and can be difficult to comprehend. Overall, the paper is well-written and contributes to the existing knowledge in its field.

We have corrected it
